# Chronotropic Response and Heart Rate Variability before and after a 160 m Walking Test in Young, Middle-Aged, Frail, and Non-Frail Older Adults

**DOI:** 10.3390/ijerph19148413

**Published:** 2022-07-09

**Authors:** Lesli Álvarez-Millán, Claudia Lerma, Daniel Castillo-Castillo, Rosa M. Quispe-Siccha, Argelia Pérez-Pacheco, Jesús Rivera-Sánchez, Ruben Fossion

**Affiliations:** 1Programa de Doctorado en Ciencias Biomédicas, Universidad Nacional Autónoma de México (UNAM), Mexico City 04510, Mexico; lesli26@ciencias.unam.mx; 2Centro de Ciencias de la Complejidad (C3), Universidad Nacional Autónoma de México (UNAM), Mexico City 04510, Mexico; 3Departamento de Instrumentación Electromecánica, Instituto Nacional de Cardiología Ignacio Chávez, Mexico City 14080, Mexico; dr.claudialerma@gmail.com; 4Servicio de Geriatría, Hospital General de México Dr. Eduardo Liceaga, Mexico City 06720, Mexico; dcastillocastillo05@gmail.com; 5Unidad de Investigación y Desarrollo Tecnológico, Hospital General de México Dr. Eduardo Liceaga, Mexico City 06720, Mexico; rosa.quispe@gmail.com (R.M.Q.-S.); argeliapp@ciencias.unam.mx (A.P.-P.); the_barbarian52@hotmail.com (J.R.-S.); 6Instituto de Ciencias Nucleares, Universidad Nacional Autónoma de México (UNAM), Mexico City 04510, Mexico

**Keywords:** early biomarkers, stimulus-response paradigm, orthostatic stressor, heart rate dynamics, autonomic response

## Abstract

The frailty syndrome is characterized by a decreased capacity to adequately respond to stressors. One of the most impaired physiological systems is the autonomous nervous system, which can be assessed through heart rate (HR) variability (HRV) analysis. In this article, we studied the chronotropic response (HR and HRV) to a walking test. We also analyzed HRV indices in rest as potential biomarkers of frailty. For this, a 160 m-walking test and two standing rest tests (before and after the walking) were performed by young (19–29 years old, *n* = 21, 57% women), middle-aged (30–59 years old, *n* = 16, 62% women), and frail older adults (>60 years old, *n* = 28, 40% women) and non-frail older adults (>60 years old, *n* = 15, 71% women), classified with the FRAIL scale and the Clinical Frailty Scale (CFS). Frequency domain parameters better allowed to distinguish between frail and non-frail older adults (low-frequency power LF, high-frequency power HF (nu), LF/HF ratio, and ECG-derived respiration rate EDR). Frail older adults showed an increased HF (nu) and EDR and a reduced LF (nu) and LF/HF compared to non-frail older adults. The increase in HF (nu) could be due to a higher breathing effort. Our results showed that a walk of 160 m is a sufficient cardiovascular stressor to exhibit an attenuated autonomic response in frail older adults. Several HRV indices showed to be potential biomarkers of frailty, being LF (nu) and the time required to reach the maximum HR the best candidates.

## 1. Introduction

The frailty syndrome associated with aging has been quantified through different approaches [1]. First, frailty may be evaluated empirically, using signs and symptoms present in the patient, using clinical scales such as the frailty phenotype [2] or the Rockwood index [3]. A second approach uses a variety of molecular or cellular biomarkers that may give evidence of alterations in physiological regulation when values are outside of the corresponding reference range [4]. Although these two approaches have been widely used to detect and understand various aspects of frailty, Varadhan et al. [1] have pointed out that these measures are “static” and fail to explain “the mechanisms underlying the vulnerability of the organism to stressors” in frail older people. On the other hand, Lipsitz [5] and Fried et al. [6] have proposed that an increased vulnerability could result from a loss of resilience in regulatory systems that help to maintain homeostasis. Therefore, Varadhan et al. [1] suggest studying the frailty syndrome in a dynamic way using a stimulus-response paradigm, as failures in regulatory systems can be subtle and undetectable in the absence of stressors [1].

The autonomous nervous system (ANS) is one of the most important regulatory systems of the human body and is fundamental to preserve homeostatic balance, particularly in situations of internal or external stress [7]. The ANS modulates cardiac activity inducing spontaneous beat-to-beat fluctuations in heart rate (HR), known as heart rate variability (HRV), which may be used as a continuous and non-invasive quantitative measure of the activity of the ANS [8].

Several previous contributions have analyzed HRV in frail, pre-frail, and non-frail older adults [9,10,11]. Most studies agree that frail older adults present failures in cardiac autonomic control, which are reflected by a low HRV as compared to non-frail and pre-frail older adults. Some studies have also documented loss of complexity in HRV time series [9]. Results for the frequency domain are contradictory; Chaves et al. [9] determined that high-frequency power (HF) in frail older adults was increased, while Katayama et al. [10] found lower HF (nu). However, these studies were carried out in different conditions, dynamic activities and resting, respectively, and may not be directly comparable. Other studies incorporated specific stimuli such as standing up, lifting the feet while sitting, or short walking tests and detected a decreased HR response in frail older adults [11,12,13].

Recent studies in frail patients have evaluated both average HR and HRV in the context of rest and walking with normal and fast speed [9,11,14]. Toosizadeh et al. [11] found that dynamic measures of HR obtained during walking showed a stronger association with frailty than measures obtained at rest. The walking test of Toosizadeh et al. [11] was realized over a short distance of 4.5 m [11], which is the standard distance used to evaluate the walking speed of older adults in the frailty-phenotype clinical scale [2]. However, evaluating longer distances, such as 400-m walking test, has been assessed to demonstrate the association between the test performance and the risk of mortality and incident cardiovascular disease in community-dwelling older adults [15].

In this study, a 160-m walking test is proposed as a long-distance test for older adults, which is much longer than the standard 4.5 m short distance test, but short enough to be completed by both non-frail and frail older adults. The aim of the study was to analyze the chronotropic response, i.e., HR response, to this 160-m walking test, and the parameters of HRV for resting and recovery after the test. To identify the effect of both age and frailty, the test was performed in groups of young, middle-aged, and older adults (frail and non-frail).

## 2. Materials and Methods

### 2.1. Study Participants

We studied a sample of 80 Mexican adults, which was divided into four groups by age and frailty status (Table 1): (i) young adults of 19–29 years old (yo) (C1, *n* = 21), (ii) middle-aged adults 30–59 yo (C2, *n* = 16), (iii) non-frail older adults >60 yo (nF, *n* = 15), and (iv) frail older adults >60 yo (F, *n* = 28). Older adults (groups nF and F) were clinically evaluated by a geriatrician who determined their frailty status according to the FRAIL scale (FS) [16] and the Clinical Frailty Scale (CFS) [3]. The FS evaluates fatigue, absence of resistance (incapacity to climb 1 flight of stairs), ambulation problems (incapacity to walk 100 m), having more than 5 illnesses, and involuntary weight loss (more than 5%) [16]. The CFS evaluates the clinical history of illnesses and a medical examination of the patient [3]. Those scales are widely used in geriatric medicine and have been validated against other known frailty scales for different populations of older adults [3] and to predict mortality [16].The patients were classified as frail when they met the criteria in both scales, patients who did not meet the criteria in both scales were classified as non-frail, and patients who met the criteria in only one of the two scales were excluded (Figure 1). Young and middle-aged adults were medical students of the Hospital General de México “Dr. Eduardo Liceaga” (HGM) and accompanying relatives of the older adults, respectively.

The study included participants of both sexes, aged 19 to 92 years, who could walk 160 m without risk of falling. Exclusion criteria were patients with Parkinson’s disease, muscular dystrophy, postural vertigo, and physical, auditory, visual, or vestibular limitation, which constitute risk factors for falls. In addition, elimination criteria were cardiac diseases where the sinoatrial node (SN) does not modulate HR, e.g., atrial fibrillation and supraventricular arrhythmias, as well as the presence of more than 5% artifacts in the ECG. We also excluded those patients taking medication that could affect the function of the SN, such as beta blockers and antiarrhythmics, e.g., Amiodarone, Diltiazem, and Digoxin. All participants signed a letter of informed consent. The study protocol was approved by the Ethics and Research Committee of the HGM under the tenets of the Declaration of Helsinki (protocol number DI/14/110-B/03/002).

### 2.2. Study Protocol

All participants were instructed to have breakfast at their home (without any food restriction) at least 1 h before the study. The walking test was performed in a quiet and isolated internal square of the surgical tower of the HGM. Before the walking test, anthropometric measures were taken, including height, weight, and body mass index (Table 1). Additionally, clinicians collected information on the clinical history of the participants, physical activity habits, and factors affecting their pulmonary capacity (smoking or have been frequently exposed to wood smoke) (Figure 1). Subsequently, a nurse placed an ambulatory physiological monitoring device (Bioharness 3.0, Zephyr Technology, Annapolis, MD, USA) on the chest of the participant, and the adequate placement of the device and the correct detection of the physiological waveforms were verified by a real-time and wireless visualization using a specialized mobile phone application (SenseView BT Zephyr Sensor for Android, Schagen, The Netherlands).

The study started with 5 min at rest in a standing position (i.e., without moving and with relaxed extended arms), which was considered the baseline phase (Figure 1). Then, each participant walked over a flat rectangular trajectory at a self-selected speed. Finally, they completed the route in one direction, and they turned around to complete the rectangle again in the other direction to cover a total distance of 160 m. A medical doctor or nurse always accompanied older adults during rest and activity. 

A distance of 160 m was a convenience distance, corresponding to twice the perimeter of the internal square where the experiment was carried out. However, this distance roughly corresponds with the validated test of 3 min (3 MWT) [17] given that older adults walk with an average gait speed of 0.89 m/s [18], and 180 s × 0.89 m/s = 160.2 m. After the walking test, participants remained 5 min at rest in a standing position, considered the recovery phase. We asked the participants to be silent during the whole test.

### 2.3. ECG Recording and Beat Detection

A chest strap (Bioharness 3.0, Zephyr Technology, Annapolis, MD, USA) was used to obtain an electrocardiogram (ECG) recording throughout the entire test, including rest and the activity phases. This device has been validated [19]. The ECG was digitized at a sampling frequency of 250 Hz and recorded in the internal memory of the device. After downloading the data on a computer, we used the open-access software Kubios HRV version 2.0 to identify the individual heartbeats in the ECG signal [20]. The Kubios software uses an automatic algorithm to detect beats, which is based on the Pan–Tompkins algorithm [21]. First, the algorithm filters the ECG signal, after which it is squared to increase the amplitude of the R waves and a moving average filter is applied. After that, a decision rule with an amplitude threshold compares the adjacent R-waves with the expected values [22]. We visually inspected the automatically generated RR heartbeat interval series for arrhythmias, ectopic beats, and artifacts, and we excluded recordings with more than 5% of arrhythmias or ectopic beats from subsequent statistical HRV analysis (Figure 1). All RR intervals were interpolated with a third-order polynomial function.

### 2.4. HRV Analysis

All HRV calculations were performed with Kubios HRV, a reliable and validated software [23,24,25,26]. The HRV analysis complied with the international recommendations [8]. Time domain, frequency domain (Table 2), and non-linear parameters (Table 3) were calculated in the 5 min interval before the walking test (baseline) and the 5 min interval after the walking test (recovery) (Figure 2). We did not analyze HRV parameters during the walking test as the series were shorter than 5 min and of different lengths for all participants. The HRV time-domain parameters included: mean heart rate (mean HR, in beats per minute), mean cardiac period (mean RR, in s), SDNN (standard deviation of all RR intervals, in ms), CV (coefficient of variation, in percentage), root mean square of the successive differences (RMSSD, in ms), pNN50 (percentage of successive RR intervals with differences larger than 50 ms), and pNN30 (percentage of successive RR intervals with differences larger than 30 ms).

In the frequency domain, the power spectrum density (PSD) was calculated for the RR intervals (Figure 1). Before the analysis, the software applied a cubic spline interpolation to the RR interval series to generate an equidistantly sampled series. The PSD was calculated with the FFT-based Welch periodogram method [28]. Frequency bands of interest were low-frequency (LF, 0.04 to 0.15 Hz) and high-frequency (HF, 0.15 to 0.4 Hz). We report LF and HF in normalized units (nu) [8]. We also included the LF/HF power ratio and the electrocardiogram-derived respiration (EDR), which estimates of the respiration frequency.

The non-linear parameters analyzed were SD1 and SD2 from the Poincaré plot (short-term and long-term variability, respectively), the SD1/SD2 ratio [29], approximate entropy (ApEn) [30], sample entropy (SampEn) [30], and the short-term scaling index α1 from Detrended Fluctuation Analysis (DFA) [31]. ApEn and SampEn measure time-series regularity and were computed using as parameters embedding dimension m = 2 and tolerance r = 0.2 × standard deviation [30]. Finally, α1 quantifies time-series self-similarity and was calculated within a short range from 4 to 11 beats [31].

### 2.5. HR Response Parameters

For the walking phase, we calculated the time required to reach the maximum HR (Δt), we also report the maximal HR (${\mathrm{HR}}_{\mathrm{Max}}$) and the velocity of HR response calculated by ${\mathrm{HR}}_{\mathrm{Max}}/\Delta t\;$.

Additionally, we calculated the chronotropic or HR response to the walking test using a parameter defined by:(1)$$\Delta {\mathrm{HR}}_{\mathrm{WB}}=\frac{{\mathrm{HR}}_{\mathrm{Walking}}-{\mathrm{HR}}_{\mathrm{baseline}}}{{\mathrm{HR}}_{\mathrm{baseline}}}\;\times \;100$$
which gives a measure in percentage of the change of HR between the walking phase and the baseline phase. Similarly, we calculated $\Delta {\mathrm{HR}}_{\mathrm{RW}}$ and $\Delta {\mathrm{HR}}_{\mathrm{RB}}$, comparing HR between recovery and walking, and recovery and baseline, respectively (Table 2).

### 2.6. Statistical Analysis

Nominal variables are described as absolute frequency and percentage and were compared between groups by a chi-squared test or an exact Fisher’s test. Most ordinal variables had no normal distribution (*p* < 0.05, Shapiro–Wilk test), in which case they were described as median (percentile 25–percentile 75) and were compared between phases (i.e., baseline versus recovery) with a Wilcoxon Rank test. An independent sample Kruskal–Wallis test was applied to compare between groups, and we applied a Mann–Whitney U test to compare the means of the pair of groups. The best cut-off point for HRV parameters was determined from receiver-operator characteristic (ROC) curve analysis, according to the shortest orthogonal distance of each curve point to the “optimal point”. The optimal point is in the coordinate (0.0,1.0). This point corresponds with maximum sensitivity and specificity. An example is shown in Figure 3. Sensitivity and specificity were estimated for each parameter. We used the Statistical Package for the Social Sciences (SPSS) version 22.0 (SPSS Inc., Chicago, IL, USA). A *p*-value ≤ 0.05 was considered significant.

## 3. Results

### Descriptive Results

By construction, age differed significantly between groups. For body mass index, there were only differences between C1 and the two groups of older adults (nF and F) (Table 1). Twelve subjects (57%) of the C1 group and eight subjects (50%) of the C2 group performed physical activity (aerobic and anaerobic) on average 1 h 3 times per week. Of the older adults, 15 subjects (100%) of the nF group performed 1 h of daily aerobic exercise. In addition, 26 subjects (93%) of the F group walked on average 800 m daily. Of the C1 Group, one subject smoked regularly and eight occasionally, whereas for the C2 group, there were three regular and one occasional smoker. Of the older adults, 10 non-frail subjects (67%) and 25 frail subjects (80%) were smokers or were exposed to wood smoke.

Figure 2 shows the HR time series during the whole test (before, during and after 160 m walking) for three selected cases of (a) a young adult, (b) a non-frail older adult, and (c) a frail adult. The young adult had an almost instantaneous chronotropic response of 20 bpm when walking, as well as an immediate HR recovery after the walking test (panel a). For the non-frail older adult, both the HR response to walking and the HR recovery after the walking test take longer than in the young adult (panel b). The basal HR of the frail older adult is higher than for the former cases and the chronotropic response and recovery were even slower than for the non-frail older adult (panel c).

Analyzing the chronotropic responses between the various rest and activity phases of the study for all participants of the young (C1), mature (C2), and non-frail (nF), and frail (F) older adults (see Table 2 and Figure 4a), we found that the HR response (ΔHR_RW_) between the recovery and walking phases, calculated with Equation (1), was negative for all the groups and significantly smaller in magnitude for the frail older adults (F) than for the young, middle-aged, and non-frail older adults (C1, C2, and nF, respectively). On the other hand, the HR response (ΔHR_RB_) between the baseline and the recovery phases, was positive and more significant in magnitude for the frail older adults (F) than for the young (C1), middle-aged (C2), and non-frail older adults (nF) where the response was negative and/or smaller in amplitude (see ΔHR_RB_ in Figure 4b).

We found that Δt as well as the speed response ($v_{\mathrm{resp}})$ were significantly higher for the frail older adults in comparison to each of the other groups (C1, C2 and nF). HR_Max_ only was different (lower) for the nF in comparison to C1 group.

Visual inspection of Figure 4a shows larger fluctuations for the young adults than for the frail and non-frail older adults in all phases (baseline, walking, and recovery), and suggests that HRV diminishes with age and even more with frailty. To study these fluctuations, we analyzed several HRV parameters, which are shown in Table 2 (time and frequency domain indices) and Table 3 (non-linear indices). First, we compared HRV indices between the recovery and baseline phases for each group (e.g., ΔpNN50 and Δα1 in Figure 4b). For group C1, we found that pNN50 was higher during recovery than in the baseline phase (Table 2). None of the frequency domain indices changed during recovery compared to the baseline for any group (Table 2). For the non-linear indices, α1 was smaller in recovery than in the baseline phase for group C1, whereas SampEn was larger during recovery than basal SampEn in group nF (Table 3).

Then, we compared the HRV indices between groups for each phase. For time domain parameters (Table 2) in the baseline phase, we found smaller SDNN, CV, RMSSD, and pNN50 for the non-frail older adults (nF) compared to the young adults (C1), we also found these parameters to be smaller for the non-frail older adults (nF) in comparison to middle aged group (C2). We found pNN30 to be smaller for non-frail older adults (nF) compared to the young adults (C1) and to be smaller for frail older adults (F) with respect to young adults (C1). SDNN was smaller for the frail older adults (F) compared to the young and mature adults (groups C1 and C2), and pNN50 was smaller for the frail older adults (F) in comparison to the young adults (C1). In contrast, pNN50 was larger in the frail group (F) compared to the non-frail group (nF). These three HRV indices (SDNN, RMSSD, and pNN50) had similar significant differences during the recovery phase, except for pNN50 between the frail (F) and non-frail (nF) groups, which was no longer significant. Further, mature adults (C2) had a smaller SDNN in comparison to young adults (C1). Moreover, there was an additional reduction of the RMSSD for frail older adults (F) compared to young adults (C1). For pNN30, the statistical differences detected during the baseline remained, and new statistical differences appeared; pNN30 was smaller for the non-frail older adults (nF) than for the mature adults (C2), and similarly for frail older adults (F) in comparison to mature adults (C2).

For the frequency-domain parameters (Table 2), during the baseline, we found that LF (nu) and the ratio LF/HF were smaller for the frail older adults (F) than for any other group (C1, C2 and nF). The opposite occurred for HF (nu), which was larger for the frail group (F). EDR is larger for the frail older adults (F) than for the non-frail older adults (nF). In the recovery phase, some of the significant differences previously found disappeared. A smaller LF (nu) and LF/HF and a larger HF (nu) prevailed in the frail (F) compared to the non-frail older adults (nF).

Table 3 shows non-linear HRV parameters. In the baseline phase, we found a smaller SD1 and SD2 in the non-frail older adults (nF) compared to young and mature adults (groups C1 and C2). SD2 was smaller for frail older adults (F) compared to young adults (groups C1 and C2). ApEn was smaller for the frail group (F) in comparison to the young adults (C1). SampEn also smaller for the frail group (F) but in comparison to mature adults (C2) and the non-frail older adults (nF). α_1_ was smaller for the frail group (F) compared to all other groups (C1, C2, nF). In the recovery phase, there were some additional differences: SD1 was smaller for the frail group (F) with respect to young adults (C1), and SD2 was smaller for the mature adults (C2) compared to the young adults (C1). SampEn was smaller for the frail older adults (F) compared to the young adults (C1) and non-frail older adults (nF). α_1_ was smaller for the frail group (F) compared to the non-frail group (nF).

Finally, we analyzed a Receiver Operating Characteristic (ROC) curve for HRV parameters that were statistically different between frail (F) and non-frail adults (nF) (Table 4). Several indices had good diagnostic values. For instance, LF (nu) and LF/HF during the baseline phase have a sensitivity of 82% (68–96%) and specificity of 71% (48–95%) and during the walking, Δt with a sensitivity of 81% (64–98%) and specificity of 70% (42–98%) (Table 5).

## 4. Discussion

### 4.1. Main Contributions of the Present Work

This work assessed the chronotropic response to a 160 m walking test in young, middle-aged, frail, and non-frail older adults. HR and HRV indices were compared between groups during the baseline or the recovery phase, showing significant differences with respect to age and/or frailty status. Moreover, comparisons between phases (recovery vs. baseline) for a specific group showed that average HR and mean RR did not recover their baseline values after the walking test in frail older adults. LF (nu), HF (nu), LF/HF, and α_1_ during the baseline and the recovery phases and SampEn during recovery phase showed (through ROC curves analysis) to be potential biomarkers to identify frailty.

### 4.2. Average HR, HR Response, and Time Domain HRV Indices

Previous studies analyzed the HR response after a stimulus in frail older adults with paradigms different from the one proposed in the present work. Romero-Ortuno et al. [12] assessed the HR response to a supine-to-standing orthostatic test, while Weiss et al. [13] evaluated a seated step test and Toosizadeh et al. [11] performed a short gait test (4.5 m). Despite the methodological differences, our results agree with Romero-Ortuño et al. [12], Weiss et al. [13], and Toosizadeh et al. [11] regarding a significantly decreased HR response to a physiological stressor in frail older adults compared to non-frail older adults and younger adults. We observed this through the ΔHR_RW_ index calculated through Equation (1) and shown in Figure 4b.

Additionally, we compared the statistical differences of HR and HRV parameters for each study group, before (baseline) and after (recovery) the 160 m walking test. Mean HR was the parameter with the most remarkable differences. The young adults group (C1) had a notable increase of the mean HR as a response to the walking test, which shows excellent adaptability to stressors; interestingly, this increase was followed by an undershoot effect of mean HR during recovery [32,33] (Figure 2a and Table 2), i.e., ΔHR_RB_ < 0. Additionally, during this phase, pNN50 showed higher parasympathetic activity and revealing not only recovery but also relaxation after the walking test in group C1. The frail older adults also showed a significant difference in mean HR from baseline to recovery phase. However, ΔHR_RB_ > 0, so that, unlike the younger group, it is not a result of their adaptability to the walking test, but instead shows their incapacity to return to their basal mean HR and showing that the walking test represented a cardiovascular stressor with an inadequate autonomic response and with diminished ability to modify the HR response. For the parameters of response to the walking test, we found that the time of the HR response (Δt) and its velocity $v_{\mathrm{resp}}$, allowed to distinguish the frail older adults from each other group (C1, C2, and nF) while the maximal HR did not. Our results shows that frail older adults did not have a significantly different maximal HR in comparison to the other groups, but their time of response was significantly higher than C1, C2, and nF groups. In addition, their velocity was significantly lower than C1, C2, and nF groups.

Analyzing the temporal parameters of HRV during the baseline phase, we compared differences between groups. We found a higher SDNN in young (C1) and middle-aged (C2) adults than in older adults (nF and F), which is consistent with the well-documented decrease of SDNN with aging [34,35]. Furthermore, a small SDNN has been associated with a higher probability or risk of frailty [9,10,11,12,13,14]. However, like Katayama et al. [10], we did not find statistical differences in the SDNN of frail and non-frail older adults. Additionally, we found a smaller RMSSD for the non-frail group (nF) than for the young and mature (groups C1 and C2). According to Toosizadeh et al. [11] and Katayama et al. [10], RMSSD is also decreased for frail older adults with respect to younger populations, although these differences were not statistically significant in the latter study. Finally, we found a larger pNN50 for young and mature adults (groups C1 and C2) than for the non-frail group. However, it was higher for the frail than for the non-frail group, contrary to Toosizadeh et al. [11], who found a larger pNN50 in the non-frail group. The implications of these results for the autonomic nervous system will be discussed later in the text.

For the recovery phase, we found very similar results as for the baseline phase, and a lower SDNN in mature adults (C2) in comparison to young adults (C1), which suggests an aging effect [34,35].

### 4.3. Frequency Domain HRV Indices

Most previous works compared frequency HRV parameters between groups during a baseline phase in rest only. In the present work, for the baseline phase, we found similar results as Chaves et al. [9] and Varadhan et al. [14]: significantly lower LF (nu) and LF/HF in frail subjects (F) compared to non-frail subjects (nF) and—in contrast—a significantly higher HF (nu) for the frail group (F) in comparison with the non-frail group (nF). These results are contrary to the findings of Katayama et al. [10], who reported higher LF (nu) and LF/HF and smaller HF (nu) in the frail group (F). Frequency-domain parameters have been related to sympathetic and parasympathetic cardiac modulation and LF/HF has been proposed as an index of sympathovagal balance [8]. The implications of these results for the autonomic nervous system will be discussed later in the text.

During the recovery phase, the differences remained only between frail (F) and non-frail groups (nF).

### 4.4. Nonlinear HRV Indices

For the nonlinear indices, during the baseline phase, we found a smaller SD1 and SD2 in non-frail older adults (nF) compared to young and mature adults (groups C1 and C2) and no statistical differences between frail (F) and non-frail older adults (nF). Toosizadeh et al. [11] reported SD1 and SD2 to be lower in frail (F) than in non-frail older adults (nF). We found a significantly smaller ApEn in frail older adults (F) compared to young adults (C1) but no difference between frail (F) and non-frail older adults (nF). Chaves et al. [9] found a smaller ApEn in frail older adults (F) than in non-frail older adults (nF), which was opposite to Takahashi et al. [36], who found a higher ApEn in frail and pre-frail than in non-frail older adults. However, Takahashi et al. [36] argues that this apparent contradiction may be explained by differences in the methodology applied, such as the timing of the recordings, the specific activities and/or stimuli applied during the test, and differences in the selected populations. We found a smaller SampEn in the frail older adults (F) compared to mature adults (C2) and non-frail older adults (nF). According to Katayama et al. [10], SampEn indeed tends to be smaller for pre-frail and frail older adults than for non-frail older adults without, however, being statistically significant. A series with a small ApEn or SampEn reflects a regular, periodic and/or predictable signal [30], which in the case of HRV data may reflect an unhealthy state that is less adaptable to stressors [37].

Finally, we found a lower α_1_ in frail older adults (F) than in all other groups (C1, C2, and nF). The α_1_ index quantifies the presence of fractal-like correlation properties; however, scaling properties of this exponent depend on posture [38]. It has been reported that α_1_ in young and middle-aged adults is on average 1.17 ± 0.05 in a standing position [39]. We found slightly higher values for young, mature, and non-frail older adults (C1, C2, and nF older adults) and a significantly lower value for frail older adults (F), α_1_ = 1.08 (0.63–1.35), which may reflect a loss of correlation in the HRV times series with frailty.

### 4.5. Pathophysiological and Clinical Implications

Stimulus-response paradigms have been proposed as a potential test that would allow to study the underlying vulnerability present in frail older adults, particularly for older adults with “hidden” frailty that only becomes apparent in the face of stressors [1]. For this study, we used a common stressor that is crucial in the daily life of older adults: walking. We proposed an extended walking test (compared to the standard distance of 4.5 m of studies on the frailty phenotype) corresponding to walking 1–2 blocks. This distance (160 m) is short enough to avoid a risk of falling among the frail older adults of our study. Only frail patients with preserved functionality (e.g., able to stand up and walk without assistance) were included in this study.

According to Toosizadeh et al. [11], walking tests increase the cardiovascular demand of the body, so these tests allow “to observe some deficits that are hidden under the basal phase”. We did not find a higher number of differences between groups during the recovery phase than in the baseline phase. However, it is essential to note that the baseline phase considered here (standing rest) is already an orthostatic stressor. Additionally, this experimental design allowed us to identify an incapacity of cardiovascular recovery of the frail older adults in an activity of daily life.

Frailty was originally linked to loss of physical functionality [2]. However, recently, frailty has been studied in a physiological context [40,41]. These new findings may be helpful in the development of early-warning biomarkers for frailty [4]. In addition, we found that HRV parameters from the frequency domain are promising candidates for the non-invasive identification of frail patients.

HRV parameters also provide physiological information. We found larger values of pNN50 and HF (nu) in the frail older adults with respect to the other populations. A higher HF (nu) is often interpreted as an increased parasympathetic modulation which is usually beneficial for health [42]. A counter example is presented by Kabbach et al. [43], who reported a higher HF in acute exacerbated patients with chronic obstructive pulmonary disease (COPD), compared to stabilized patients with COPD. They suggest that the increase in this HRV parameter is not a sign of good health but rather a consequence of a vagal influence on the airways as exacerbated patients make a higher effort to breathe, possibly due to bronchoconstriction [43]. We observed a higher breathing rate (EDR) in frail older adults than in non-frail older adults. Therefore, we suggest that the higher HF (nu) in frail older adults could be due to an increased breathing rate rather than an increased parasympathetic modulation [42]. We cannot interpret the dominance or absence of sympathetic modulation based on the LF components as it has been found that LF incorporates information of the two modulations, both sympathetic and parasympathetic [44], therefore, to understand specifically the alterations of the sympathetic modulation in frail older adults, a different experimental protocol should be carried out. The results found in this article emphasize the importance of comparing baseline phases of rest with phases with specific stimuli to allow a better interpretation of the sympathetic, parasympathetic, and respiratory responses.

### 4.6. Perspectives

HRV analysis with advanced methods such as machine learning are being explored in diverse applications [45] including classification of frail and non-frail patients [46,47]. Our approach in this first exploratory study was to apply traditional statistical tests to identify HRV indices that had different median values when comparing frail and non-frail patients. Then, we explored the potential value for classification of frail and non-frail by calculating sensitivity and specificity using the basic ROC curve analysis. We selected a candidate for cut-off value by the simplest method: calculating the minimum orthogonal distance between the optimum point of the ROC curve (0,1) and each point within the ROC curve. This simple approach was enough to show the potential value of HRV analysis during a 160 m walking test to identify frail patients based on a physiological measurement (the cardiac response to the test). These results, based on a small sample of participants are enough to warrant further studies in larger populations where machine learning methods would improve the classification of frail and non-frail patients.

### 4.7. Study Limitations

For the walking test, we selected a fixed distance of 160 m which participants covered at a self-selected speed in 3 min or less. However, for HRV analysis, a minimum RR time series length of 5 min is recommended [8]. Therefore, we only analyzed HRV in the baseline and recovery phases. Additionally, the duration of the walking phase is different for each subject. The study included a relatively small sample of participants (*n* = 80), which reduces the statistical power of the study. Although the internal validity was strengthened by carefully applied selection criteria and assessment procedures, further studies are needed to increase the external validity, by assessing larger and more heterogeneous samples of both frail and non-frail older adults. A high prevalence of comorbidities in older adults could have a confusing effect on the results of this study; however, we excluded diseases that direct affect HRV. In future works, we plan to explore the HRV data with other mathematical methods such as recurrence plots and analyses of cardiorespiratory coupling.

## 5. Conclusions

The chronotropic response to the 160 m walking test comparing baseline to recovery showed that frail older adults did not recover their basal HR, suggesting that walking a moderate distance of 160 m at a self-selected speed constitutes a more important cardiovascular stressor for the frail older adults than for young, middle-aged, and non-frail older adults. It is important to note that 160 m is a distance commonly walked in the daily routine of the older adults (e.g., walking to a shop).

Additionally, our findings highlight the importance of studying in detail HRV parameters in frail and non-frail older adults, as they help to understand physiological impairments of the ANS that may be characteristic of the frailty syndrome. We studied several parameters of the temporal, frequency, and non-linear domains and found that the best parameters to distinguish between frail and non-frail older adults are the frequency domain parameters. Notably, HF (nu) was higher and LF (nu) and LF/HF were lower in the frail older adults relative to the non-frail older adults and/or young or middle-aged adults. HR series were more predictable (smaller SampEn) in the frail older adults and through the analysis of α1, we found indications for a loss of correlations in the HRV time series. Furthermore, HRV parameters showed to be potential biomarkers for frailty based on non-invasive methods. It is important to recall that frail older adults are a widely heterogeneous population with several and different diseases. Therefore, further studies are required to validate the present results in more extensive populations.

## Figures and Tables

**Figure 1 ijerph-19-08413-f001:**
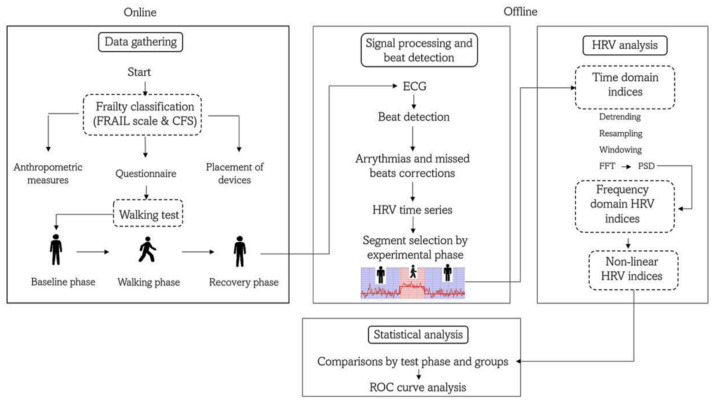
Flow diagram of the study protocol. FRAIL = fatigue, resistance, ambulation, illness, and low performance scale; CFS = clinical frailty scale; HRV = heart rate variability; FFT = Fast Fourier Transform; PSD = power spectrum density; ROC = receiver operator characteristic.

**Figure 2 ijerph-19-08413-f002:**
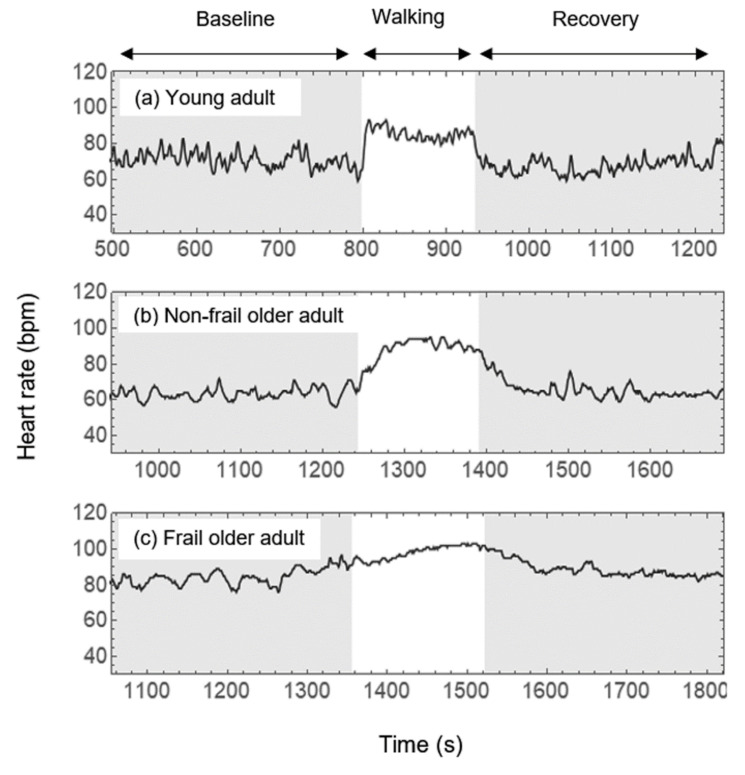
Heart rate time series from a selected (**a**) young adult, (**b**) a non-frail older adult and (**c**) a frail older adult during the whole test. The left fragment shaded in gray corresponds to the first 5 min of standing rest (baseline). The white part shows the fragment where participants walked a fixed distance of 160 m. Fragment sizes of walking are different as every person walked at their own preferred pace. The right gray shaded fragment corresponds with the recovery phase, which means 5 min of standing rest. Adapted with permission from Ref. [27]. 2022, AIP Publishing.

**Figure 3 ijerph-19-08413-f003:**
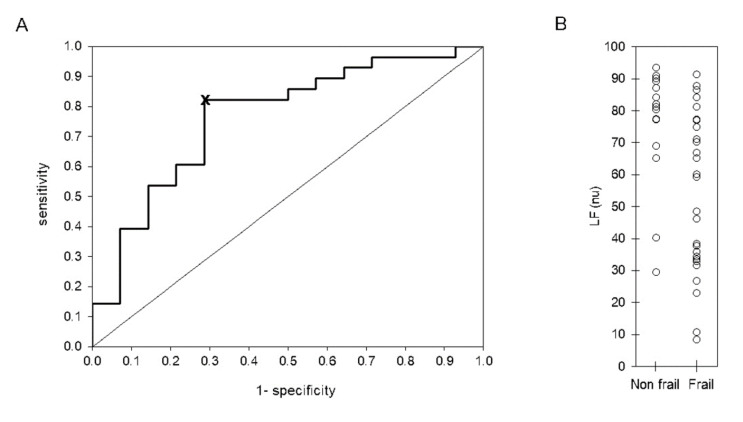
Receiver-operator characteristic (ROC) curve (**A**) and scatter plot (**B**) of LF (nu) evaluated during 5 min of standing rest (baseline phase) in 15 non-frail and 27 frail older adults. The best cut-off point, indicated by symbol ‘x’ in panel (**A**), was determined according to the shortest orthogonal distance from each point to the optimum vale (0,1). The line in panel (**B**) indicates the cut-off point at 77.3 normalized units (nu).

**Figure 4 ijerph-19-08413-f004:**
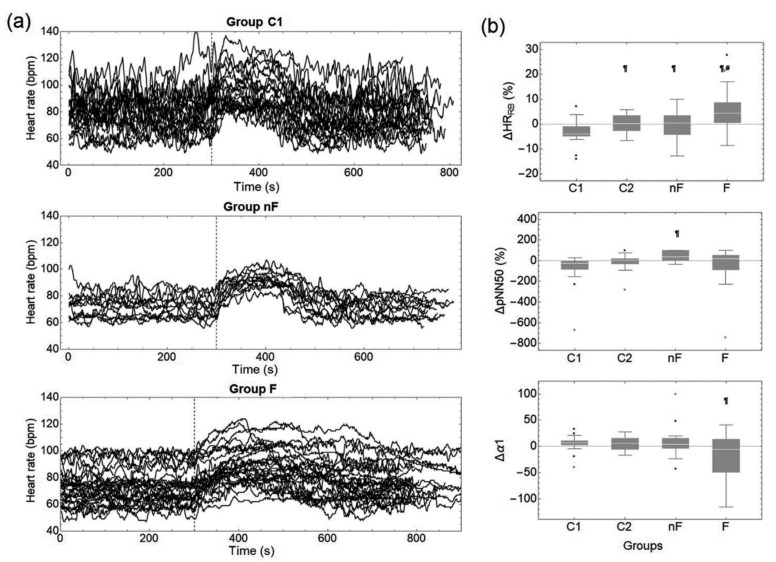
(**a**) Heart rate time series from young adults (group C1), non-frail older adults (group nF) and frail older adults (group F) during the whole test. Dashed vertical line represent the final of baseline phase (first 5 min). The final of the walking test cannot be specified as every subject walked to their own pace and traveled the fixed distance of 160 m in different times. (**b**) Chronotropic response ΔHR and changes in selected HRV parameters comparing recovery vs. baseline phases expressed as a percentage. Changes were calculated through Equation (1) for young (C1), middle-aged adults (C2), non-frail older adults (nF), and frail adults (F). Black dots in panel (**b**) represent the outliers of the data. ^¶^ *p* < 0.05 compared to C1 (comparison between groups). ^#^ *p* < 0.05 compared to C1 (comparison between groups).

**Table 1 ijerph-19-08413-t001:** Demographic and most common comorbidities of the study participants. Results are reported as mean ± standard deviation or absolute value (percentage).

Variables	C1	C2	nF	F
	(*n* = 21)	(*n* = 16)	(*n* = 15)	(*n* = 28)
Age (years)	22.0 ± 1.4	48.6 ± 8.9 *	71.5 ± 8.2 *^,^**	78 ± 5.4 *^,^**^,||^
Sex				
Male	9 (43%)	6 (38%)	9 (60%)	8 (29%)
Female	12 (57%)	10 (62%)	6 (40%)	20 (71%)
Body mass index (kg/m^2^)	23.8 ± 3.6	27.5 ± 5.9	27.4 ± 3.6 *	26.4 ± 3.9 *
Physical activity				
Mild	9 (43%)	8 (50%)	0 (0%)	2 (7%)
Moderate	12 (57%)	8 (50%)	6 (40%)	20 (71%)
Comorbidities				
Hypertension	0	3	4	15
Mild cognitive	0	0	0	14
Diabetes	0	0	2	10
Depression	0	0	2	9

* *p* < 0.05 compared to C1; ** *p* < 0.05 compared to C2; || *p* < 0.05 compared to nF.

**Table 2 ijerph-19-08413-t002:** Heart rate response ΔHR and linear HRV parameters evaluated from a 160-m walking test in young adults (C1), middle-aged adults (C2), non-frail older adults (nF), and frail older adults.

	C1 (*n* = 21)	C2 (*n* = 16)	nF (*n* = 15)	F (*n* = 28)	*p*-Value
* **Baseline** *					
HR (1/min) (bpm)	81.2 (69.9–89.9)	74.2 (65.1–84.4)	76.4 (65.9–82.5)	74.1 (67.2–84.5)	0.462
Mean RR (ms)	747 (672–866)	811 (713–924)	788 (736–912)	811 (721–908)	0.540
SDNN (ms)	61.5 (46.7–76.4)	48.6 (35.3–63.8)	29.6 (27.6–46.7) **^,||^	33.2 (20.6–57.2) **^,||^	0.000 ^#^
CV (%)	82.5 (53.8–98.6)	68.5 (46.8–102.0)	42.9 (33.1–68.1) **^,||^	46.0 (26.5–88.6) **	0.007 ^#^
RMSSD (ms)	33.5 (25.5–42.1)	27.9 (17.3–39.5)	13.5 (7.7–25.7) **^,||^	22.4 (9.7–71.8)	0.010 ^#^
pNN30 (%)	34.3 (24.6–45.1)	22.4 (13.7–35.9)	4.6 (1.2–32.6) **	3.7 (7.6–26.6) **	0.006 ^#^
pNN50 (%)	12 (3.00–22.00)	6 (1.00–16.00)	1 (0.00–7.00) **^,||^	2 (0.00–12.00) **^,¶^	0.004 ^#^
LF (ms^2^)	1202.5 (1066.2–1842.6)	531.1 (473.5–1464.4) **	172.9 (74.5–916.0) **^,||^	167.8 (53.6–1041.0) **^,||^	0.000 ^#^
HF (ms^2^)	380.5 (247.1–1263.6)	275.7 (176.9–617.7)	67.5 (38.5–201.1) **^,||^	97.4 (186.8–2130.0) **	0.002 ^#^
LF (nu)	75.5 (64–85.5)	74.0 (57.5–87.9)	81.3 (69.0–89.3)	59.8 (34.0–77.1) **^,||,¶^	0.011 ^#^
HF (nu)	24.0 (14.4–35.8)	26.0 (12.0–42.4)	18.6 (10.7–31.0)	40.1 (22.9–64.9) **^,||,¶^	0.010 ^#^
LF/HF	3.14 (1.79–5.93)	2.84 (1.35–7.33)	4.38 (2.23–8.38)	1.49 (0.52–3.37) **^,||,¶^	0.010 ^#^
EDR (Hz)	0.2 (0.16–0.21)	0.18 (0.17–0.19)	0.17 (0.14–0.19)	0.19 (0.16–0.22) ^¶^	0.214
* **Recovery** *					
HR (1/min)	77.2 (67.00–88.2) *	75.5 (61.7–86.2)	75.2 (66.0–79.5)	78.6 (69.1–87.6) *	0.582
Mean RR (ms)	786 (685–900) *	811 (700–983)	801 (757–914)	766 (694–873) *	0.619
SDNN (ms)	59.1 (52.6–82.4)	51.2 (42.4–61.7) **	29.7 (19.1–37.2) **^,||^	28.0 (21.9–49.6) **^,||^	0.000 ^#^
CV (%)	75.5 (58.8–122.3)	68.5 (49.0–96.4)	39.6 (24.5–58.0) **^,||^	41.1 (24.6–72.3) **^,||^	0.000 ^#^
RMSSD (ms)	37.9 (29.7–46.1)	35.1 (14.4–39.0)	15.0 (12.5–21.9) **^,||^	16.5 (8.3–58.7) **	0.003 ^#^
pNN30 (%)	39.7 (30.5–51.5)	30.7 (17.5–44.8)	4.4 (1.7–18.2) **^,||^	2.8 (5.4–20.9) **^,||^	0.000 ^#^
pNN50 (%)	16.00 (10.00–26.00) *	12.00 (1.00–20.00)	0.00 (0.00–2.00) **^,||^	1.00 (0.00–22.00) **	0.000 ^#^
LF (ms^2^)	1564.1 (1234.8–2595.7)	916.5 (617.3–1414.6) **	362.7 (180.6–646.5) **^,||^	138.1 (144.6–950.7) **^,||^	0.000 ^#^
HF (ms^2^)	601.2 (482.0–1301.4)	319.9 (221.5–651.6)	77.0 (21.9–259.9) **^,||^	65.3 (189.5–1066.9) **	0.001 ^#^
LF (nu)	73.3 (60.4–80.1)	77.0 (45.1–89.4)	81.5 (68.5–86.5)	68.8 (42.6–77.6) ^¶^	0.098
HF (nu)	26.7 (19.6–39.5)	23.0 (10.6–54.9)	18.4 (13.4–31.1)	31.3 (22.3–57.2) ^¶^	0.101
LF/HF	2.74 (1.53–4.08)	3.35 (0.82–8.43)	4.49 (2.20–6.44)	2.19 (0.75–3.47) ^¶^	0.102
EDR (Hz)	0.19 (0.16–0.23)	0.17 (0.16–0.19)	0.17 (0.13–0.19)	0.18 (0.16–0.22)	0.431
HR ***response***					
$\Delta t\;\left(s\right)$	39.0 (33.1–48.2)	42 (29.8–59.6))	54 (26.3–84.7)	115 (96.5–175.3) **^,||,¶^	0.000 ^#^
${\mathrm{HR}}_{\mathrm{Max}}\;\left(\mathrm{bpm}\right)$	107.0 (101.4–114.4)	104.2 (91.2–107.9)	98.3 (90.9–103.4) **	88.3 (87.1–100.2)	0.013 ^#^
$v_{\mathrm{resp}}\;\left(\mathrm{bpm}/s\right)$	2.4 (2.3–4.1)	2.5 (1.5–5.1)	1.8 (1.0–6.7)	0.7 (0.6–1.9) **^,||,¶^	0.000 ^#^
$\Delta {\mathrm{HR}}_{\mathrm{WB}}$ (%)	20.0 (12.2– 24.9)	16.5 (−11.3–28.6)	16.4 (1.3–26.2)	15.2 (8.6–23.1)	0.734
$\Delta {\mathrm{HR}}_{\mathrm{RW}\;}$(%)	−19.5 (−12.7–−23.3)	−15.9 (−10.1–−25.4)	−16.2 (−11.9–−26.2)	−8.8 (−5.0–36.9) **^,||,¶^	0.002 ^#^
$\Delta {\mathrm{HR}}_{\mathrm{RB}}$ (%)	−3.9 (−4.9–−0.9)	0.1 (−2.8–3.4)	0.4 (−4.3–3.4)	4.3 (0.5–8.6) **^,||,¶^	0.001 ^#^

*n*: number of participants. Data is shown as median (percentile 25–percentile 75). ^#^ *p* < 0.05 for Kruskal–Wallis test. * *p* < 0.05 compared to baseline (comparison between phases for the same group). ** *p* < 0.05 compared to C1 (comparison between groups for the same phase). ^||^ *p* < 0.05 compared to C2 (comparison between groups for the same phase). ^¶^ *p* < 0.05 compared to nF (comparison between groups for the same phase).

**Table 3 ijerph-19-08413-t003:** Non-linear HRV parameters evaluated from a 160-m walking test in young adults (C1), middle-aged adults (C2), non-frail older adults (nF), and frail older adults. *n*: number of participants. Data is shown as median (percentile 25 –percentile 75).

	C1 (*n* = 21)	C2 (*n* = 16)	Nf (*n* = 15)	F (*n* = 28)	*p*-Value
* **Baseline** *					
SD1 (ms)	23.7 (18.0–29.8)	19.8 (12.3–28.0)	9.5 (5.4–18.2) **^,||^	15.9 (6.9–50.8)	0.010 ^#^
SD2 (ms)	81.7 (64.3–104.1)	67.0 (48.4–87.6)	39.8 (38.4–63.1) **^,||^	39.8 (27.1–73.7) **^,||^	0.000 ^#^
SD1/SD2	0.3 (0.2–0.4)	0.3 (0.2–0.4)	0.2 (0.2–0.3)	0.3 (0.2–0.8)	0.136
ApEn	1.11 (1.03–1.19)	1.07 (1.03–1.13)	1.04 (0.97–1.14)	1.02 (0.95–1.11) **	0.149
SampEn	1.35 (1.18–1.63)	1.27 (1.11–1.64)	1.18 (1.03–1.51)	1.16 (0.96–1.39) ^||,¶^	0.353
α_1_	1.37 (1.16–1.47)	1.40 (1.06–1.66)	1.41 (1.26–1.57)	1.08 (0.63–1.35) **^,||,¶^	0.006 ^#^
* **Recovery** *					
SD1 (ms)	26.8 (21.0–32.7)	24.9 (10.2–27.6)	10.6 (8.9–15.5) **^,||^	11.7 (5.8–41.6) **	0.003 ^#^
SD2 (ms)	79.8 (68.8–109.9)	71.7 (54.6–74.4) **	40.5 (26.5–51.5) **^,||^	38.5 (30.1–61.6) **^,||^	0.000 ^#^
SD1/SD2	0.3 (0.2–0.4)	0.3 (0.2–0.5)	0.3 (0.2–0.4)	0.3 (0.2–0.8)	0.381
ApEn	1.12 (0.97–1.17)	1.06 (0.92–1.12)	1.12 (1.01–1.18)	1.02 (0.91–1.13)	0.184
SampEn	1.52 (1.25–1.67)	1.22 (0.94–1.71)	1.39 (1.14–1.69) *	1.21 (0.96–1.37) **^,¶^	0.065
α_1_	1.27 (1.09–1.37) *	1.40 (0.88–1.54)	1.37 (1.18–1.51)	1.17 (0.79–1.32) ^¶^	0.075

^#^ *p* < 0.05 for Kruskal–Wallis test. * *p* < 0.05 compared to baseline (comparison between phases for the same group). ** *p* < 0.05 compared to C1 (comparison between groups for the same phase). ^||^ *p* < 0.05 compared to C2 (comparison between groups for the same phase). ^¶^ *p* < 0.05 compared to nF (comparison between groups for the same phase).

**Table 4 ijerph-19-08413-t004:** ROC curve analysis of heart rate response (ΔHR) and heart rate variability indices.

	AUC (95% CI)	*p*-Value	Best Cut off Value
* **Baseline** *			
pNN50 (%)	0.583 (0.408–0.758)	0.386	0.8
LF (nu)	0.765 (0.609–0.922)	0.080	77.3
HF (nu)	0.768 (0.612–0.924)	0.005	22.6
LF/HF	0.768 (0.612–0.924)	0.005	3.4
EDR (Hz)	0.686 (0.526–0.847)	0.051	0.2
SampEn (beats)	0.597 (0.420–0.774)	0.090	1.2
α_1_	0.745 (0.891–0.599)	0.010	1.2
* **Recovery** *			
LF (nu)	0.722 (0.554–0.890)	0.020	78.6
HF (nu)	0.722 (0.554–0.890)	0.020	21.4
LF/HF	0.722 (0.554–0.890)	0.020	3.7
SampEn	0.724 (0.564–0.885)	0.019	1.3
α_1_	0.717 (0.557–0.876)	0.023	1.2
HR ***response***			
Δt (s)	0.807 (0.652–0.962)	0.006	74
$v_{\mathrm{resp}}\;$(bpm/s)	0.824 (0.678–0.970)	0.000	1.2
ΔHR_RB_	0.707 (0.544–0.870)	0.083	−3.8

**Table 5 ijerph-19-08413-t005:** Predictive values of HRV indexes in 28 frail and 15 non-frail older adults.

	Sensitivity (%)	Specificity (%)	PPV (%)	NPV (%)
* **Baseline** *				
pNN50 > 0.8%	43 (25–61)	43 (17–69)	60 (39–81)	27 (9–46)
LF < 77.3 nu	82 (68–96)	71 (48–95)	85 (72–99)	67 (43–91)
HF >22.6 nu	18 (4–32)	29 (5–52)	33 (9–57)	15 (1–28)
LF/HF > 3.4	82 (68–96)	71 (48–95)	85 (72–99)	67 (43–91)
EDR (Hz) > 0.2	36 (18–53)	36 (11–61)	53 (30–75)	22 (5–39)
SampEn > 1.2	54 (35–72)	64 (39–89)	75 (56–94)	41 (20–61)
α_1_ > 1.2	61 (43–79)	86 (67–104)	89 (76–103)	52 (32–73)
* **Recovery** *				
LF < 78.6 nu	79 (63–94)	57 (31–83)	79 (63–94)	57 (31–83)
HF > 21.3 nu	21 (6–37)	43 (17–69)	43 (17–69)	21 (6–37)
LF/HF > 3.7	79 (63–94)	57 (31–83)	79 (63–94)	57 (31–83)
SampEn > 1.3	71 (55–88)	64 (39–89)	80 (64–96)	53 (29–77)
α_1_ > 1.2	64 (47–82)	79 (57–100)	86 (71–101)	52 (31–74)
HR ***response***				
Δt (s)	81 (64–98)	70 (42–98)	85 (69–101)	64 (35–92)
$v_{\mathrm{resp}}\;$(bpm/s)	76 (58–94)	70 (42–98)	84 (68–101)	58 (30–86)
Δ_HR_ > −3.6	54 (35–72)	86 (67–104)	88 (73–104)	48 (28–68)

## Data Availability

The data presented in this study are available on request from the corresponding author.

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
