# Peer review of "Chronotropic Response and Heart Rate Variability before and after a 160 m Walking Test in Young, Middle-Aged, Frail, and Non-Frail Older Adults"

_ijerph, 2022, doi:10.3390/ijerph19148413_

Round 1

Reviewer 1 Report

The manuscript presents chronotropic response to a walking test in subjects of different ages, additionally dividing older participants into frail and non-frail ones.  It is well written and the objectives of the study are reflected in the results and conclusions. Its main limitation is the low sample size, which significantly reduces the statistical power of the study - it should be emphasized in the text. A few more issues require further clarification:

  1. The last sentences of the introduction containing the study findings should be removed to the results section.
  2. Did the authors exclude from the study patients using drugs that affect the sinoatrial node - this should be clarified.
  3. The authors should include a brief psychometric description of the FRAIL scale and Clinical Frailty Scale (CFS) in the methods section.
  4. Statistical analysis - comparisons between the four groups should include ANOVA test. In this section the sentence "Wilcoxon Rank test and within groups with a Mann- Whitney U test" (lines 178-179 ) should relate to comparisons between the two groups. However, in the results section, e.g. the sentence ,, For the temporal parameters (Table 2) in the baseline phase, we found smaller SDNN, CV, RMSSD and 269 pNN50 for the non-frail older adults compared to young and middle-aged adults (C1 and C2) (lines 268-271)" refers to the comparison of the three tested subgroups - this issue needs to be clarified. 

Author Response

Manuscript ijerph-1702329 Chronotropic response and heart rate variability before and after a 160m walking test in young, middle-aged, frail, and non-frail older adults

Response to Reviewer 1

Comment: The manuscript presents chronotropic response to a walking test in subjects of different ages, additionally dividing older participants into frail and non-frail ones.  It is well written and the objectives of the study are reflected in the results and conclusions. Its main limitation is the low sample size, which significantly reduces the statistical power of the study - it should be emphasized in the text.

Response: Thank you for your comments, which helped us to improve our manuscript. Now we emphasize the study limitation regarding the reduced statistical power (lines 547-549).

Comment: A few more issues require further clarification:

  1. The last sentences of the introduction containing the study findings should be removed to the results section.

Response: The last sentences were removed from the Introduction. To avoid redundant description of the findings we did not moved the sentences to the results section.

Comment: 2. Did the authors exclude from the study patients using drugs that affect the sinoatrial node - this should be clarified.

Response: In the section of methodology, we now specify the following (lines 117-119).

Comment: 3. The authors should include a brief psychometric description of the FRAIL scale and Clinical Frailty Scale (CFS) in the methods section.

Response: We extended the description of FRAIL and CFS scales as follows (lines 100-105).

Comment: 4. Statistical analysis - comparisons between the four groups should include ANOVA test. In this section the sentence "Wilcoxon Rank test and within groups with a Mann- Whitney U test" (lines 178-179 ) should relate to comparisons between the two groups.

Response: We applied a Kruskal Wallis test for independent samples, which is the equivalent to ANOVA for non-parametric distributions, and is now included in Tables 2 and 3: (lines 324-340).

Comment: However, in the results section, e.g. the sentence ,, For the temporal parameters (Table 2) in the baseline phase, we found smaller SDNN, CV, RMSSD and 269 pNN50 for the non-frail older adults compared to young and middle-aged adults (C1 and C2) (lines 268-271)" refers to the comparison of the three tested subgroups - this issue needs to be clarified. 

Response: Your observation is correct, we clarified redaction in lines (342-345).

Reviewer 2 Report

The authors investigate heart rate variability (HRV) before (baseline) and after (recovery) a 160m walking test in frail and non-frail elderly subjects and in two healthy control groups (young and older adults). They find that mean RR is the only parameter that changes significantly between baseline and recovery in frail elderly subjects. During recovery, frequency domain parameters LF (nu), HF (nu) and LF/HF showed differences between the frail and non-frail elderly subjects. Furthermore, SampEn and DFA alpha1 differed between these two groups during baseline and recovery. Moreover, the heart rate response (relative difference between recovery and baseline) was clearly different for the frail elderly subjects compared to the other groups. The authors conclude that especially the heart rate response to the 160m walking test may be used to test for frailty in elderly subjects. Furthermore, HRV parameters should be further investigated to gain more insight in the alterations in cardiac autonomic regulation in this condition.

Comments:

I enjoyed reading the manuscript. I just have some comments:

Abstract:

The authors should add information at the beginning of the abstract with respect to the background of the study. Furthermore, they should add information on the characteristics of the groups (e.g. number of subjects, age, gender, FRAIL state)

l. 59, statement “… while Katayama et al. [10] found lower.” I guess it should be “… while Katayama et al. [10] found lower HF.” Please clarify.

l. 110, statement “Additionally, clinician collect ed information…” probably should be “Additionally, clinicians collected information”. Please clarify

Methods

l. 143 calculation of pNN50

The results of pNN50 (c.f. table 2) indicate that the threshold of 50 ms might by too and, hence, pNN50 is rather low. Obviously, the variations in the RR tachogram are not large enough during standing and walking. Did the authors try e.g. pNN30 to obtain more meaningful results? (see also: Mietus, J. E., et al. (2002). "The pNNx files: re-examining a widely used heart rate variability measure." Heart 88(4): 378-380.)

For the sake of completeness, please report also the absolute values of LF and HF. These values add complementary information to LF (nu) and HF(nu) and give an impression of how large these oscillations are.

l. 169, Figure 1 clearly shows that also time plays a crucial role in the heart rate response between baseline -> walking and walking -> recovery. Healthy young subjects show an almost immediate response whereas the response in the elderly subjects is clearly slower. Hence, I suggest to calculate some kind of measure that reflects the speed of the response, e. g. Delta HR / delta t. This some more relevant information is captured from the data. This ‘speed’ can be calculated for the transition baseline -> walking and walking -> recovery.

l. 180 statement “according to the shortest orthogonal distance of each curve point to the optimal point.” For those not familiar with ROC: the ‘optimal point’ (0.0, 1.0) should be at least stated oer briefly explained.

l. 268 statement “For the temporal parameters…” I guess it should be “For the time domain parameters…” Please clarify.

l. 299, Table 5: I guess there’s a mistake for LF/HF. According to table 4 it should be LF/HF>3.4 instead of LF/HF>77.3. Please clarify.

l. 349ff. The authors discuss the results of LF and HF. However, it is not clear whether the cited references also use LF (nu) and HF (nu) or whether they state LF and HF in absolute units. Please clarify.

Author Response

Manuscript ijerph-1702329 Chronotropic response and heart rate variability before and after a 160m walking test in young, middle-aged, frail, and non-frail older adults

Response to Reviewer 2

The authors investigate heart rate variability (HRV) before (baseline) and after (recovery) a 160m walking test in frail and non-frail elderly subjects and in two healthy control groups (young and older adults). They find that mean RR is the only parameter that changes significantly between baseline and recovery in frail elderly subjects. During recovery, frequency domain parameters LF (nu), HF (nu) and LF/HF showed differences between the frail and non-frail elderly subjects. Furthermore, SampEn and DFA alpha1 differed between these two groups during baseline and recovery. Moreover, the heart rate response (relative difference between recovery and baseline) was clearly different for the frail elderly subjects compared to the other groups. The authors conclude that especially the heart rate response to the 160m walking test may be used to test for frailty in elderly subjects. Furthermore, HRV parameters should be further investigated to gain more insight in the alterations in cardiac autonomic regulation in this condition.

Comments:

Comment: I enjoyed reading the manuscript. I just have some comments:

 Response: Thank you for your comments, which helped us to improve our manuscript.

Abstract:

Comment: The authors should add information at the beginning of the abstract with respect to the background of the study. Furthermore, they should add information on the characteristics of the groups (e.g. number of subjects, age, gender, FRAIL state)

 Response: We added the requested background and specific information to the abstract (lines 17 to 25):

Comment: l. 59, statement “… while Katayama et al. [10] found lower.” I guess it should be “… while Katayama et al. [10] found lower HF.” Please clarify.

 Response: Your observation is correct. We made the correction on line 66.

Comment: l. 110, statement “Additionally, clinician collect ed information…” probably should be “Additionally, clinicians collected information”. Please clarify

Response: Your observation is correct. We made the correction on line 133. 

Methods

Comment:l. 143 calculation of pNN50. The results of pNN50 (c.f. table 2) indicate that the threshold of 50 ms might by too and, hence, pNN50 is rather low. Obviously, the variations in the RR tachogram are not large enough during standing and walking. Did the authors try e.g. pNN30 to obtain more meaningful results? (see also: Mietus, J. E., et al. (2002). "The pNNx files: re-examining a widely used heart rate variability measure." Heart 88(4): 378-380.)

Response: We calculated pNN30 and added the results in Table 2. More significant differences were found between the frail group and the young (C1) or middle-aged adults (C2), but not between frail and non-frail older adults. These results are now described on lines 346-348 and 356-359.

Comment: For the sake of completeness, please report also the absolute values of LF and HF. These values add complementary information to LF (nu) and HF(nu) and give an impression of how large these oscillations are.

 Response: The results of LF and HF in absolute values are now included on Table 2 (line 324).

Comment: l. 169, Figure 1 clearly shows that also time plays a crucial role in the heart rate response between baseline -> walking and walking -> recovery. Healthy young subjects show an almost immediate response whereas the response in the elderly subjects is clearly slower. Hence, I suggest to calculate some kind of measure that reflects the speed of the response, e. g. Delta HR / delta t. This some more relevant information is captured from the data. This ‘speed’ can be calculated for the transition baseline -> walking and walking -> recovery.

 Response: Thank you for this insightful suggestion. We calculated the proposed measures of maximum heart rate, delta t and the speed response (Table 2). We found that both delta t and the speed response were different between frail and non-frail patients, and both had a significant area under the ROC curve (Table 4) with good predictive values (Table 5). (We mentioned the parameters in lines 221, 301-303, 423-428).

Comment: l. 180 statement “according to the shortest orthogonal distance of each curve point to the optimal point.” For those not familiar with ROC: the ‘optimal point’ (0.0, 1.0) should be at least stated oer briefly explained.

 Response: We now specified that the ‘optimal point’ is in the coordinates (0.0,1.0) and that it corresponds with maximal sensibility and specificity (line 241-242).

Comment: l. 268 statement “For the temporal parameters…” I guess it should be “For the time domain parameters…” Please clarify.

 Response: Your observation is correct. We have made the change on line 342.

Comment: l. 299, Table 5: I guess there’s a mistake for LF/HF. According to table 4 it should be LF/HF>3.4 instead of LF/HF>77.3. Please clarify.

 Response: Your observation is correct, we have already changed the value in table 5.

Comment: l. 349ff. The authors discuss the results of LF and HF. However, it is not clear whether the cited references also use LF (nu) and HF (nu) or whether they state LF and HF in absolute units. Please clarify.

Response: We clarified the units (lines 450-454).

Reviewer 3 Report

Please find the attachment

Author Response

We thank the reviewer for his/her helpful comments which helped to improve our manuscript. Please find all point by point responses in the attached PDF.

Round 2

Reviewer 1 Report

The authors made appropriate corrections. I have no additional comments. 

Reviewer 3 Report

The authors have addressed most of the comments. This manuscript can be accepted after minor English editing. Authors MUST improve the figures' resolution quality before acceptance.